# Differences between Acute Exacerbations of Idiopathic Pulmonary Fibrosis and Other Interstitial Lung Diseases

**DOI:** 10.3390/diagnostics11091623

**Published:** 2021-09-06

**Authors:** Paola Faverio, Anna Stainer, Sara Conti, Fabiana Madotto, Federica De Giacomi, Matteo Della Zoppa, Ada Vancheri, Maria Rosaria Pellegrino, Roberto Tonelli, Stefania Cerri, Enrico M. Clini, Lorenzo Giovanni Mantovani, Alberto Pesci, Fabrizio Luppi

**Affiliations:** 1Respiratory Unit, Cardio-Thoracic-Vascular Department, University of Milano Bicocca, San Gerardo Hospital, ASST Monza, 20900 Monza, Italy; paola.faverio@unimib.it (P.F.); annetta.stainer@gmail.com (A.S.); i.fede@live.it (F.D.G.); matteo.dellazoppa@gmail.com (M.D.Z.); alberto.pesci@unimib.it (A.P.); 2Department of Health Science, Università degli Studi di Milano-Bicocca, 20900 Monza, Italy; sara.conti@unimib.it (S.C.); lorenzo.mantovani@unimib.it (L.G.M.); 3Value-Based Healthcare Unit, IRCCS Multimedica, 20099 Sesto San Giovanni, Milan, Italy; fabiana.madotto@unimib.it; 4Regional Referral Centre for Rare Lung Disease, Department of Clinical and Experimental Medicine, University of Catania, 95030 Catania, Italy; adact1@hotmail.it; 5Centre for Rare Lung Diseases, University Hospital of Modena, 41124 Modena, Italy; mr.pellegry@gmail.com (M.R.P.); roberto.tonelli@me.com (R.T.); stefania.cerri@unimore.it (S.C.); enrico.clini@unimore.it (E.M.C.); 6Department of Medicine and Surgery, University of Milan Bicocca, 20900 Monza, Italy

**Keywords:** acute exacerbation, idiopathic pulmonary fibrosis, progressive fibrosing interstitial lung disease, fibrosing lung diseases

## Abstract

Interstitial lung diseases (ILDs) comprise a wide group of pulmonary parenchymal disorders. These patients may experience acute respiratory deteriorations of their respiratory condition, termed “acute exacerbation” (AE). The incidence of AE-ILD seems to be lower than idiopathic pulmonary fibrosis (IPF), but prognosis and prognostic factors are largely unrecognized. We retrospectively analyzed a cohort of 158 consecutive adult patients hospitalized for AE-ILD in two Italian university hospitals from 2009 to 2016. Patients included in the analysis were divided into two groups: non-IPF (62%) and IPF (38%). Among ILDs included in the non-IPF group, the most frequent diagnoses were non-specific interstitial pneumonia (NSIP) (42%) and connective tissue disease (CTD)-ILD (20%). Mortality during hospitalization was significantly different between the two groups: 19% in the non-IPF group and 43% in the IPF group. AEs of ILDs are difficult-to-predict events and are burdened by relevant mortality. Increased inflammatory markers, such as neutrophilia on the differential blood cell count (HR 1.02 (CI 1.01–1.04)), the presence of pulmonary hypertension (HR 1.85 (CI 1.17–2.92)), and the diagnosis of IPF (HR 2.31 (CI 1.55–3.46)), resulted in negative prognostic factors in our analysis. Otherwise, lymphocytosis on the differential count seemed to act as a protective prognostic factor (OR 0.938 (CI 0.884–0.995)). Further prospective, large-scale, real-world data are needed to support and confirm the impact of our findings.

## 1. Introduction

Interstitial lung diseases (ILDs) comprise a wide group of pulmonary parenchymal disorders that are classified together because of similarities in their clinical presentation, chest radiographic appearance, and physiologic features. ILDs include idiopathic pulmonary fibrosis (IPF), hypersensitivity pneumonitis (HP), connective tissue disease-related ILD (CTD-ILD), and idiopathic non-specific interstitial pneumonia (iNSIP) [1]. Although the prototype of progressive ILD is considered IPF [2], an important subset of patients with other fibrotic ILDs experience a decline in lung function with progressive symptoms, poor response to treatment, and reduced quality of life [3]. However, patients with ILDs may experience acute respiratory deteriorations, termed “acute exacerbation” (AE) [4]. Most of the studies available focused on IPF and showed an annual incidence of AE ranging between 5 and 19%, which varied across different studies according to the statistical methodology, differences in the study populations, and the definition of AE [4,5,6,7]. Prognosis is poor with high mortality rates and a median survival of less than one year [6]. Prognostic factors include a detection of an UIP pattern, the severity of hypoxemia, and the alteration of pre-exacerbation lung functional parameters [6]. While AEs of IPF (AE-IPF) have been widely studied, clinical data on AE-ILDs other than IPF are limited [4,8]. The incidence of AE-ILD seems to be lower than IPF, but prognosis and prognostic factors are largely unrecognized [8,9]. 

The aim of our study was to describe the patients’ characteristics at hospitalization for AE-ILD, comparing IPF with non-IPF patients, in two Italian ILD referral centers (University Hospitals of Monza and Modena), and to identify predictors of death at hospital discharge and of overall survival.

## 2. Material and Methods

This was a retrospective cohort study of consecutive adult patients hospitalized for AE-ILD in two Italian University Hospitals (San Gerardo Hospital, Monza, and Policlinico di Modena, Modena) from 1 October 2009 to 31 December 2016. Medical records in the period of interest were initially included if at least one of the International Classification of Diseases, Ninth Revision, Clinical Modification (ICD-9-CM) codes reported in Figure 1 were met. In the absence of a specific code for AE-ILDs, only patients with both diagnoses of ILD and respiratory failure present during the same hospital admission were included. Of the 4894 hospitalizations included, 174 subjects met all the criteria for AE; however, reviewing on a case-by-case basis all medical records, 16 patients had more than one hospitalization and were excluded from the analysis. One hundred fifty-eight patients were included for final analysis (Figure 1). 

The type of ILD was re-evaluated using the current international criteria [1,10]. AE-ILDs were defined according to the 2007 definition by Collard et al. as a subjective worsening of dyspnea within the month prior to presentation and new ground glass opacities or consolidation at chest imaging after exclusion of alternative diagnosis including congestive heart failure (CHF), pulmonary embolism (PE), pulmonary infections, and pneumothorax [11]. 

Active pulmonary infections were identified by the physician in charge based on patients’ signs and symptoms and laboratory and microbiological analysis results. Nosocomial infections developed during hospitalization for AE were not excluded from the study. 

Radiological findings on admission, both chest X-rays and CT scans, were reviewed during a multidisciplinary discussion to confirm/exclude the diagnosis of AE. 

During each hospitalization, we collected information on patient’s characteristics, symptoms, physical examination, blood tests, pre-existing comorbidities, and therapies used before and during the hospitalization from the medical record. Nevertheless, we searched electronic databases of the two centers for the last available pre-hospitalization spirometry, and we included pulmonary function tests performed within 12 months before the AE. Patients were followed-up until 31 January 2018 to assess their vital status and the date of death.

We carried out descriptive analyses for all collected variables. We then assessed factors associated with mortality at discharge using univariate and multivariate logistic models, with death at discharge as the outcome of interest. The best multivariate model was selected based on a stepwise selection process. Thereafter, we analyzed the survival time from hospitalization, and we built overall Kaplan–Meier curves, which were stratified for patients with and without IPF. Based on these curves, we estimated the probability of survival at one, three, and six months as well as at one year after the AE. Finally, we assessed survival predictors through univariate and multivariate Cox proportional hazard models. The proportional hazard assumption was tested through the analysis of Schoenfeld’s residuals, but also by using graphical methods. The best multivariate model was once again selected based on a stepwise selection process. The threshold for statistical significance was *p* < 0.05. The outcomes of interest were in-hospital mortality and all-cause mortality on 31 January 2018. When transplanted, patients were excluded from the follow-up evaluation.

## 3. Results

We retrospectively identified 4894 patients, accounting for 423 unique hospitalizations for acute respiratory failure. Of these, we collected 158 patients, 121 (69%) of which were admitted at San Gerardo Hospital, Monza, and 53 (31%) of which were admitted at the University Hospital of Modena meeting the inclusion and exclusion criteria for AE-ILDs (IPF and non-IPF). These patients were included in the analysis and divided into two groups: non-IPF (98 patients—62%) and IPF (60 patients—38%). Among ILDs included in the non-IPF group, the most frequent diagnoses were NSIP (42%) and CTD-ILD (20%). (Figure 2).

Compared with non-IPF patients, those with IPF were predominantly males (81% vs. 59%—*p* 0.0019) and former or ex-smokers (82% vs. 53%—*p* 0.004). The age at diagnosis and at first exacerbation were similar between the two groups. No seasonal variations in incidence were noted in the whole cohort and between the two groups (Table 1).

Comorbidities were detected in 92% of patients at the baseline. The most common were systemic hypertension (46%), diabetes (26%), heart failure and coronary artery disease (30%), and gastroesophageal reflux (25%). However, lung cancer was significantly associated with IPF, while other forms of cancer were significantly associated with the non-IPF (11% vs. 0%—*p* < 0.001 and 17% vs. 0%—*p* < 0.001, respectively). 

Lung function tests were performed during clinical stability before the AE, showing that patients with AE had a moderate restrictive respiratory impairment (a mean forced vital capacity (FVC) = 64% (standard deviation (STD) = 18.69) of the predicted value, mean total lung capacity (TLC) = 65% (STD = 17.17) of the predicted value), with a lower mean FVC and FVC% predicted in the non-IPF compared to the IPF group (1.77 L (STD = 0.7) vs. 2.04 L (STD = 0.79) and 62% (STD = 19.1) vs. 66% (STD = 17.9), respectively (Table 2)).

The mean duration of AEs was 14 days, without significant differences between the two groups. The most common symptoms during AEs were dyspnea (97%), cough (36%), and fever (24%). Symptoms and physical examination did not vary significantly between the two groups. The presence of bilateral inspiratory crackles at chest auscultation was the most common sign and was significantly more frequent in the IPF group (94% vs. 77%—*p* 0.003). 

Blood tests at hospital admission showed increased inflammatory markers in both groups. Moreover, C-reactive protein (CRP) was more frequently increased in the non-IPF group (78% vs. 58%—*p* 0.007), but patients in the IPF group showed higher CRP median values (3.64 (interquartile range (IQR) 1.23–5.19) vs. 1.42 (IQR 0.51–4.1)—*p* 0.02). Mild leukocytosis with neutrophilia was detected in both groups; however, neutrophilia was more pronounced in patients with IPF compared with non-IPF patients (Table 1)**.** During hospitalization for AE, patients were treated with different therapies according to the clinical experience of our ILD team, without any significant difference between the two groups, with the exception of antibiotic therapy, which was prescribed more frequently in patients with IPF. Oxygen therapy was administered in 89%, systemic steroids in 71%, and immunosuppressive therapy in 14% of patients. Regarding ventilation, 32% of patients underwent non-invasive ventilation during hospitalization, and 6% of patients underwent invasive ventilation, with no significant differences between the two groups (Table 3).

Mortality during hospitalization was statistically different between the two groups: 19% in the non-IPF group and 43% in the IPF group (*p* 0.0007) (Figure 3). 

Mortality at 1, 3, 6, and 12 months was estimated using the Kaplan–Meier survival curve from the first AE. The overall survival of the cohort was 75% (CI 67.59–81.17) at one month, 57% (CI 49.01–64.48) at three months, 51% (CI 43.28–58.88) at six months, and 46% (CI 37.65–53.18) at one year. The non-IPF group showed a better survival compared to the IPF group at every point in time. 

On the multivariate logistic model addressing the probability of death at discharge, a higher white blood cell (WBC) count at the baseline was significantly associated with an increased risk (odds ratio (OR) 1.089 (confidence interval (CI) 1.011–1.174)). At the same time, lymphocytosis on the differential cell count was shown to be protective (OR 0.938 (CI 0.884–0.995)). Furthermore, on the multivariate Cox proportional hazard model, the presence of neutrophilia and pulmonary hypertension (PH) resulted in a risk factor for mortality (HR 1.02 (CI 1.01–1.04) and 1.85 (CI 1.17–2.92), respectively). 

The IPF diagnosis was confirmed as a risk factor for death in both analyses (OR 2.95 (CI 1.026–8.335); HR 2.31 (CI 1.55–3.46)). Missing values are reported in Table 4.

## 4. Discussion

All 158 patients included in our study experienced acute respiratory decline with new or worsening ground-glass opacities and/or consolidation superimposed on a background pattern of fibrotic lung disease. After the exclusion of any other identifiable cause of pulmonary damage, according to the 2007 consensus statement criteria by Collard et. al. [11], our cohort was divided in two groups: non-IPF and IPF. It is known that an AE can occur not only in IPF but also in virtually all ILDs, including rare ILDs (e.g., pleuro-parenchimal fibroelastosis). In the non-IPF group, AE were more frequently detected in INSIP (42%) and CTD-ILD (20%). However, other studies reported different incidence of ILD subtypes, with a higher prevalence of AE of CTD-ILD in their cohorts [9,12,13]. Compared with non-IPF patients, patients with IPF were predominantly males (81% vs. 59%-p 0.0019), whereas AE-ILD had a similar incidence in males and females (64% vs. 45%). While this result is in line with data from other studies in AE of IPF [5], AE of ILDs other than IPF have been reported more frequently in women [14]. The mean age at AE in our study was similar between the two groups, a finding recently confirmed by Suzuki et al. [9]. However, in other studies, the mean age at AE seemed to be lower in non-IPF patients [14,15]. Some studies highlighted a seasonal variation in AE incidence, with an increased frequency during winter and spring [16], but we did not find any difference in our study. A different proportion of ILD subtypes in the non-IPF population and a different study design could explain these differences. 

At the time of presentation, patients in both groups were on different therapeutic regimens, including oral corticosteroids associated or not with cytotoxic agents. Only a minority of patients with IPF were receiving pirfenidone or nintedanib. Most patients experienced AE prior to the routine use of these two antifibrotic drugs. It is reasonable to expect that the number of AEs in these patients will decrease in the future according to their ability to reduce disease progression and their possible protective role against AE [5,9,16]. A recent retrospective study on the AE of chronic fibrosing interstitial pneumonia in patients receiving antifibrotic agents has shown estimated one-, two-, and three-year AE incidence rates of 11.4%, 32%, and 36.3%, respectively [17].

There are no evidence-based therapeutic strategies for the management of AE, and the treatment does not differ between the two groups; all patients were treated with high-flow oxygen, a high dose of intravenous corticosteroids, and broad-spectrum antibiotics. Immunosuppressant therapy was administered only in 7 patients in the IPF and in 16 in the non-IPF group. Treatment does not differ in the literature [5,18].

Mortality during hospitalization was statistically different between the two groups (19% in the non-IPF group and 43% in the IPF group), although no differences in the treatment could justify this data. Moreover, IPF patients showed a worse survival at every time point when mortality was estimated (at 1, 3, 6, and 12 months). A comparison between various types of ILDs (including IPF and non-IPF patients) has already been described in the literature [13,15,19,20]. Nevertheless, some authors described similar outcomes between IPF and non-IPF patients [9,21]. A different study design and the inclusion of different subtypes of ILDs could explain this difference. In our study, a multivariate logistic model addressing the probability of death and Cox proportional hazard models of survival highlighted multiple risk factors for mortality. 

Both IPF and non-IPF patients presented an increase in inflammatory markers. Furthermore, patients with IPF experiencing an AE were characterized by higher median values of CRP and more prominent neutrophilic leukocytosis. On multivariate analysis, a higher WBC count at the baseline and the presence of neutrophilia resulted in risk factors for mortality. On the contrary, the presence of lymphocytosis on the differential cell count was shown to be protective. 

The pathogenesis of AE for IPF and other ILDs remains unknown. Previous research on IPF suggests that intrinsic factors, as the progression of the underlying disease or an aggravation caused by external factors or both, could trigger AE [22]. Several studies support the hypothesis of a role for inflammation in the pathogenesis of AE. Song et al. identified elevated values of CRP as an independent risk factor for AE [23]. Another study on AE of IPF identified the Glasgow Prognostic Score (GPS), an inflammation-based prognostic scoring system based on albumin and CRP levels, as a potential predictive factor of mortality in IPF patients experiencing an AE [24]. Usui et al., in a cohort of patients with AE of fibrosing ILD including IPF, showed that slightly increased procalcitonin and the presence of a systemic inflammatory response syndrome (SIRS) were independent risk factors for mortality [25]. Moreover, a recent meta-analysis confirmed that higher WBC counts and lactate dehydrogenases (LDH) values are prognostic factors of AE for IPF [26]. 

An increased monocyte count has been shown to be a prognostic marker in pulmonary fibrosis and other fibrotic disorders [27,28]. Moreover, Kawamura et al. demonstrated that the absolute monocyte count is an independent risk factor for AE in patients with fibrosing ILDs [17]. However, we did not find any difference in the monocyte count in our cohort. 

In our study, non-IPF patients with AE showed a better survival compared with patients with IPF. It could be reasonably supposed that AEs of ILDs other than IPF slightly differ in their pathogenetic mechanism from IPF, being characterized by lower levels of inflammation and a better prognosis. 

Another significant risk factor for death at first AE on the multivariate Cox proportional hazard model was the presence of pulmonary hypertension. This finding has been confirmed in a recent systematic review and meta-analysis of risk factors for AE in IPF [29].

Our study shows several limitations. First, data were collected retrospectively and could present associated biases, although this was minimized by the inclusion of consecutive patients. Second, the study implementation was limited to two hospitals and therefore included a relatively small number of cohort cases; the lack of statistical significance of some findings may be due to insufficient power.

Third, the patients included in the study were selected based on the 2007 definition of AEs, since our cohort was collected before the new definition was available. However, the new definition of AE did not include non-IPF ILD subtypes. The lack of a proper definition for AE of non-IPF ILDs in the literature justifies our choice. This definition was modified in 2016 to include known causes of DAD, but this modification has not been incorporated into the other ILD subtypes.

Fourth, the majority of IPF patients were not on antifibrotic drugs when they experienced AE. The use of antifibrotic drugs was not routine clinical practice during the follow-up period considered in our study; therefore, the incidence of AE could be lower today. 

Fifth, several patients were on systemic steroids or immunosuppressive drugs before hospitalization, and the values of WBC and the differential cell count could have been influenced by therapy.

## 5. Conclusions

AEs of ILDs are difficult-to-predict events and are burdened by the relevant mortality. Increased inflammatory markers with neutrophilia on the differential blood cell count, the presence of PH, and the diagnosis of IPF resulted as negative prognostic factors, while lymphocytosis on the differential blood cell count seemed to act as a protective prognostic factor.

Further prospective, large-scale, real-world data are needed to support and confirm the impact of our findings.

## Figures and Tables

**Figure 1 diagnostics-11-01623-f001:**
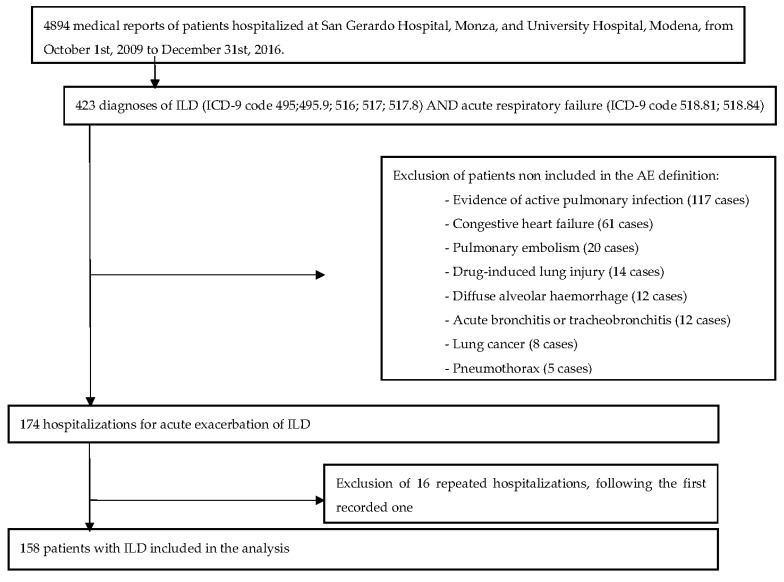
Flowchart of the study.

**Figure 2 diagnostics-11-01623-f002:**
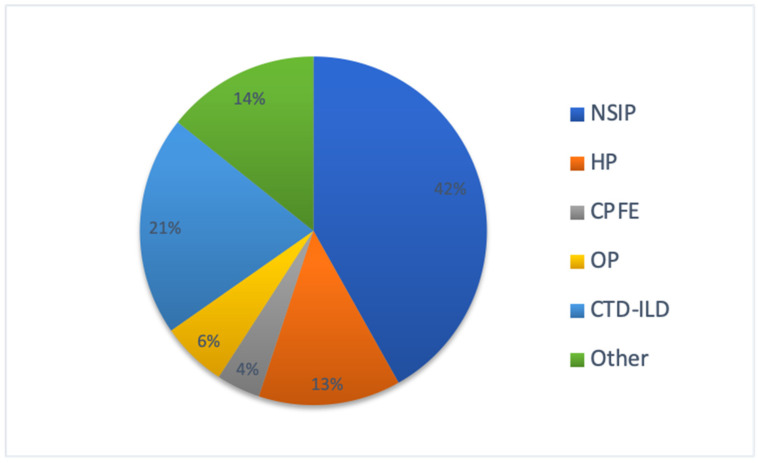
Distribution of different ILDs in the non-IPF group. NSIP: non-specific interstitial pneumonia; HP: hypersensitivity pneumonia; CPFE: combined pulmonary fibrosis and emphysema; OP: organizing pneumonia; CTD-ILD: connective-tissue-disease-related ILD. CTD-ILD included: Sjogren syndrome, mixed connectivitis, systemic sclerosis, rheumatoid arthritis, antisynthetase syndrome, p-ANCA vasculitis. Others included: undifferentiated ILD, AIP, LIP, AFOP, and eosinophilic pneumonia.

**Figure 3 diagnostics-11-01623-f003:**
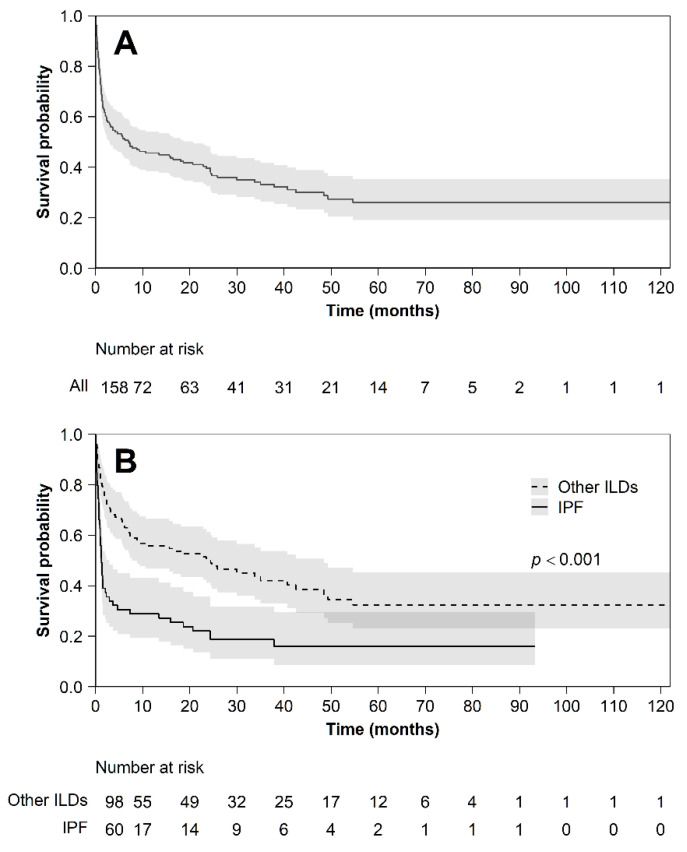
Kaplan–Meier survival curve from the first AE. Panel (**A**) whole cohort; panel (**B**) stratified between subjects with IPF and other ILDs.

**Table 1 diagnostics-11-01623-t001:** Baseline characteristics, clinical presentation, and blood exams at hospital admission.

	IPF		Total (N = 158)	*p*-Value
	No (N = 98)	Yes (N = 60)		
**Baseline characteristics**				
Men—N (%)	58 (59.18)	48 (80.00)	106 (67.09)	0.0069
Age at exacerbation—median (Q1–Q3)	70 (62–75)	70 (65–76.5)	70 (64–76)	0.1930
Age at diagnosis—median (Q1–Q3)	67.5 (59–74)	68 (60–74)	68 (60–74)	0.7972
Duration PF—median (Q1–Q3)	9 (0–36)	24 (8–48)	12 (1–43)	0.0072
Smoker—N (%)				0.0003
Never	40 (40.82)	11 (18.33)	51 (32.28)	
Current/ex	52 (53.06)	48 (80.00)	100 (63.29)	
Comorbidities—N (%)				
Lung cancer	0 (0.00)	7 (11.67)	7 (4.43)	<0.0001
Other cancers	17 (17.35)	0 (0.00)	17 (10.76)	<0.0001
Arterial hypertension	47 (47.96)	25 (41.67)	72 (45.57)	0.4408
Diabetes	23 (23.47)	15 (25.00)	38 (24.05)	0.8271
Tachyarrhythmia	15 (15.31)	9 (15.00)	24 (15.19)	0.9374
Heart failure and/or coronary artery disease	26 (26.53)	20 (33.33)	46 (29.11)	0.3610
Gastro-esophageal reflux disease	30 (30.61)	11 (18.33)	41 (25.95)	0.0875
Osteoporosis	15 (15.31)	8 (13.33)	23 (14.56)	0.7137
Hypothyroidism	8 (8.16)	5 (8.33)	13 (8.23)	1.0000
Pulmonary hypertension	21 (21.43)	13 (21.67)	34 (21.52)	0.9718
Anxious depressive syndrome	12 (12.24)	3 (5.00)	15 (9.49)	0.1316
**Therapies before hospitalization**				
Pirfenidone—N (%)	2 (2.04)	12 (20.00)	14 (8.86)	0.0001
Nintedanib—N (%)	1 (1.02)	1 (1.67)	2 (1.27)	1.0000
**Clinical presentation at admission**				
**Symptoms**				
Chest pain—N (%)	4 (4.08)	3 (5.00)	7 (4.43)	1.0000
Dyspnea—N (%)	95 (96.94)	59 (98.33)	154 (97.47)	0.2918
Cough—N (%)	40 (40.82)	16 (26.67)	56 (35.44)	0.0827
Fever—N (%)	28 (28.57)	12 (20.00)	40 (25.32)	0.2516
Sputum—N (%)				0.5435
No	75 (76.53)	51 (85.00)	126 (79.75)	
Clear	13 (13.27)	6 (10.00)	19 (12.03)	
Mucoid	2 (2.04)	2 (3.33)	4 (2.53)	
Mucopurulent	3 (3.06)	0 (0.00)	3 (1.90)	
Bloody	2 (2.04)	0 (0.00)	2 (1.27)	
Duration (days)—median (Q1–Q3)	10 (3–30)	7 (4–15)	10 (3–25)	0.3415
**Physical examination**				
Bilateral inspiratory crackles—N (%)	74 (75.51)	56 (93.33)	130 (82.28)	0.0003
Reduced vesicular murmur—N (%)	18 (18.37)	8 (13.33)	26 (16.46)	0.7313
Peripheral edema—N (%)	17 (17.35)	12 (20.00)	29 (18.35)	0.5944
**Blood exams at admission**				
White blood cell count (10^3^/µL)—median (Q1–Q3)	9.91 (7.25–13.53)	10.9 (8.9–13.6)	10.28 (7.59–13.53)	0.1129
Neutrophilia—N (%)				0.0283
Normal	48 (48.98)	19 (31.67)	67 (42.41)	
Pathological	49 (50.00)	41 (68.33)	90 (56.96)	
Neutrophilia (10^3^/µL)—median (Q1–Q3)	7.18 (4.68–10.78)	8.90 (6.18–11.00)	8.09 (5.40–10.88)	0.0372
Neutrophilia %—median (Q1–Q3)	75.4 (64.3–85.2)	80.3 (68.9–86.7)	77 (65.4–85.9)	0.2200
Lymphocytes (10^3^/µL)—median (Q1–Q3)	1.34 (0.96–1.93)	1.4 (1–2.03)	1.39 (0.99–1.97)	0.2127
Lymphocytes %—median (Q1–Q3)	14.5 (9.6–23.25)	16.1 (9.7–20.2)	14.6 (9.6–22.1)	0.8728
Monocytes (10^3^/µL)—median (Q1–Q3)	0.63 (0.45–0.85)	0.68 (0.5–0.8)	0.65 (0.49–0.85)	0.6019
Monocytes %—median (Q1–Q3)	6.55 (4.55–9)	6 (4.3–7.9)	6.3 (4.5–8.3)	0.2486
Eosinophils (10^3^/µL)—median (Q1–Q3)	0.13 (0.03–0.3)	0.06 (0.03–0.18)	0.1 (0.03–0.28)	0.1572
Eosinophils %—median (Q1–Q3)	1.35 (0.2–3.2)	0.7 (0.2–1.5)	0.9 (0.2–2.7)	0.0663
Basophils (10^3^/µL)—median (Q1–Q3)	0.02 (0.01–0.04)	0.02 (0.02–0.03)	0.02 (0.01–0.04)	0.8109
Basophils %—median (Q1–Q3)	0.2 (0.1–0.4)	0.2 (0.1–0.3)	0.2 (0.1–0.4)	0.3359
CRP-N(%)				0.0072
Normal	25 (25.51)	5 (8.33)	30 (18.99)	
Pathological	71 (72.45)	54 (90.00)	125 (79.11)	
CRP (mg/dL)—median (Q1–Q3)	1.18 (0.49–4.06)	3.55 (1.29–5)	1.8 (0.57–4.8)	0.0123

Most of the variables had no missing values, and among the other variables the highest percentage of missing values was recorded for reduced vesicular murmur and symptoms duration (20.9%). A thorough description of missing values is reported in Table 4. Footnotes: STD—standard deviation; PF—pulmonary fibrosis; IPF—idiopathic pulmonary fibrosis; CRP—C-reactive protein. Comparison between the IPF and the non-IPF groups using Chi-square, Fisher’s exact test, *T*-test, or Mann—Whitney U-test for independent samples as appropriate.

**Table 2 diagnostics-11-01623-t002:** Lung function tests in stability.

	IPF		Total (N = 158)	*p*-Value
	No (N = 98)	Yes (N = 60)		
FVC (L)—median (Q1–Q3)	1.86 (1.18–2.35)	1.98 (1.46–2.48)	1.895 (1.26–2.36)	0.1010
FVC %—median (Q1–Q3)	62 (49–74)	69 (53–82)	62 (50–79)	0.2432
FEV1 (L)—median (Q1–Q3)	1.59 (1.04–1.98)	1.78 (1.43–2.19)	1.69 (1.19–2.09)	0.0063
FEV1 %—median (Q1–Q3)	68 (56–86)	78.775 (61–91)	72 (57–88)	0.0707
DLCO (mmol/min/kPa)—median (Q1–Q3)	2.48 (1.32–3.88)	3.32 (1.88–7.09)	2.91 (1.74–6.00)	0.0752
DLCO %—median (Q1–Q3)	36 (23–45)	27 (20–38)	32 (22- 42)	0.1260
TLC (L)—median (Q1–Q3)	3.49 (2.79–4.44)	3.53 (3.03–4.22)	3.50 (2.82–4.42)	0.6133
TLC %—median (Q1–Q3)	64 (52–76)	65.4 (50–76)	64 (52–76)	0.6665

Missing values ranged from 19.0% for FVC to 53.2% for DLCO. A thorough description of missing values is reported in Table 4. Footnotes: STD—standard deviation; IPF—idiopathic pulmonary fibrosis; FEV1—forced expiratory volume in the 1st second; FVC—forced vital capacity; DLCO—diffusing capacity of the lung for carbon monoxide; TLC—total lung capacity. Comparison between the IPF and the non-IPF groups using Chi-square or Mann–Whitney U-test for independent samples as appropriate.

**Table 3 diagnostics-11-01623-t003:** Therapy during the hospitalization for AE and outcomes.

	IPF		Total (N = 158)	*p*-Value
	No (N = 98)	Yes (N = 60)		
**Therapies during hospitalization**				
Oxygen supplementation—N (%)	87 (88.78)	56 (93.33)	143 (90.51)	0.3727
Anticoagulant—N (%)	48 (48.98)	32 (53.33)	80 (50.63)	0.4622
Intravenous steroids—N (%)	66 (67.35)	48 (80.00)	114 (72.15)	0.0566
Antibiotics—N (%)	62 (63.27)	48 (80.00)	110 (69.62)	0.0165
Immunosuppressives—N (%)				0.3756
Cyclophosphamide	11 (11.22)	6 (10.00)	17 (10.76)	
Azathioprine	3 (3.06)	0 (0.00)	3 (1.90)	
Methotrexate	0 (0.00)	1 (1.67)	1 (0.63)	
NIV during hospitalization—N (%)	35 (35.71)	15 (25.00)	50 (31.65)	0.1801
IMV during hospitalization—N (%)	8 (8.16)	3 (5.00)	11 (6.96)	0.4642
**Hospitalization length (days)—median (Q1–Q3)**	12 (8–18)	14.5 (9–23)	13 (9–20)	0.1539
**In-hospital death—N (%)**	21 (21.43)	28 (46.67)	49 (31.01)	0.0003

Footnotes: IPF—idiopathic pulmonary fibrosis; NIV—non-invasive mechanical ventilation; IMV—invasive mechanical ventilation. Comparison between the IPF and the non-IPF groups using Chi-square or Mann–Whitney U-test for independent samples as appropriate.

**Table 4 diagnostics-11-01623-t004:** Distribution of missing values for all analysed variables.

	IPF		Total
	No	Yes	
**Baseline characteristics**			
Gender	0 (0.00)	0 (0.00)	0 (0.00)
Diagnosis	0 (0.00)	0 (0.00)	0 (0.00)
Age at exacerbation	0 (0.00)	0 (0.00)	0 (0.00)
Age at diagnosis	4 (4.08)	9 (15.00)	13 (8.23)
Duration PF	1 (1.02)	10 (16.67)	11 (6.96)
Smoker	6 (6.12)	1 (1.67)	7 (4.43)
**Comorbidities**			
Lung cancer	0 (0.00)	0 (0.00)	0 (0.00)
Other cancers	0 (0.00)	0 (0.00)	0 (0.00)
Arterial hypertension	0 (0.00)	0 (0.00)	0 (0.00)
Diabetes	0 (0.00)	0 (0.00)	0 (0.00)
Tachyarrhythmia	1 (1.02)	0 (0.00)	1 (0.63)
Heart failure and/or coronary artery disease	0 (0.00)	0 (0.00)	0 (0.00)
Gastro-esophageal reflux disease	0 (0.00)	0 (0.00)	0 (0.00)
Osteoporosis	1 (1.02)	0 (0.00)	1 (0.63)
Hypothyroidism	0 (0.00)	0 (0.00)	0 (0.00)
Pulmonary hypertension	0 (0.00)	0 (0.00)	0 (0.00)
Anxious depressive syndrome	0 (0.00)	0 (0.00)	0 (0.00)
**Therapies before hospitalization**			
Pirfenidone	1 (1.02)	0 (0.00)	1 (0.63)
Nintedanib	1 (1.02)	0 (0.00)	1 (0.63)
**Clinical presentation at admission**			
**Symptoms**			
Chest pain	1 (1.02)	1 (1.67)	2 (1.27)
Dyspnea	0 (0.00)	1 (1.67)	1 (0.63)
Cough	0 (0.00)	1 (1.67)	1 (0.63)
Fever	0 (0.00)	1 (1.67)	1 (0.63)
Sputum	3 (3.06)	1 (1.67)	4 (2.53)
Duration (days)	18 (18.37)	15 (25.00)	33 (20.89)
**Physical examination**			
Bilateral inspiratory crackles	1 (1.02)	3 (5.00)	4 (2.53)
Reduced vesicular murmur	15 (15.31)	18 (30.00)	33 (20.89)
Peripheral edema	3 (3.06)	4 (6.67)	7 (4.43)
**Blood exams at admission**			
White blood cell count 10^3^/µL	3 (3.06)	3 (5.00)	6 (3.80)
Neutrophilia	1 (1.02)	0 (0.00)	1 (0.63)
Neutrophilia 10^3^/µl	6 (6.12)	3 (5.00)	9 (5.70)
Neutrophilia %	6 (6.12)	3 (5.00)	9 (5.70)
Lymphocytes 10^3^/µl	6 (6.12)	7 (11.67)	13 (8.23)
Lymphocytes %	6 (6.12)	7 (11.67)	13 (8.23)
Monocytes 10^3^/µl	6 (6.12)	7 (11.67)	13 (8.23)
Monocytes %	6 (6.12)	7 (11.67)	13 (8.23)
Eosinophils 10^3^/µl	6 (6.12)	7 (11.67)	13 (8.23)
Eosinophils %	6 (6.12)	7 (11.67)	13 (8.23)
Basophils 10^3^/µl	6 (6.12)	7 (11.67)	13 (8.23)
Basophils %	6 (6.12)	7 (11.67)	13 (8.23)
CRP	2 (2.04)	1 (1.67)	3 (1.90)
CRP mg/dL	7 (7.14)	13 (21.67)	20 (12.66)
**Lung function tests in stability**			
FVC (L)	19 (19.39)	11 (18.33)	30 (18.99)
FVC %	19 (19.39)	11 (18.33)	30 (18.99)
FEV1(L)	25 (25.51)	12 (20.00)	37 (23.42)
FEV1 %	27 (27.55)	12 (20.00)	39 (24.68)
DLCO (mmoL/min/kPa)	60 (61.22)	24 (40.00)	84 (53.16)
DLCO %	43 (43.88)	19 (31.67)	62 (39.24)
TLC (L)	33 (33.67)	14 (23.33)	47 (29.75)
TLC %	34 (34.69)	15 (25.00)	49 (31.01)
**Therapies during hospitalization**			
Oxygen supplementation	1 (1.02)	1 (1.67)	2 (1.27)
Anticoagulants	2 (2.04)	3 (5.00)	5 (3.16)
Intravenous steroids	0 (0.00)	1 (1.67)	1 (0.63)
Antibiotics	0 (0.00)	1 (1.67)	1 (0.63)
Immunosuppressives	1 (1.02)	0 (0.00)	1 (0.63)
NIV during hospitalization	0 (0.00)	1 (1.67)	1 (0.63)
IMV during hospitalization	0 (0.00)	1 (1.67)	1 (0.63)
**Hospitalization length, days**	0 (0.00)	0 (0.00)	0 (0.00)
**In-hospital death**	0 (0.00)	0 (0.00)	0 (0.00)

## Data Availability

Data are available upon request.

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
