# Peer review of "Differences between Acute Exacerbations of Idiopathic Pulmonary Fibrosis and Other Interstitial Lung Diseases"

_diagnostics, 2021, doi:10.3390/diagnostics11091623_

Round 1

Reviewer 1 Report

I have read the article by Faverio et al. with great interest. The authors compared IPF and non-IPF exacerbations of patients with ILD. This is a valuable work which should be endorsed for publication, should the authors address some comments.

  • I understand that this is a retrospective study, but post-hoc sensitivity analyses should be performed if the sample size was large enough to deliver the conclusions.
  • Numerous comparisons were performed between the IPF and non-IPF groups. Did the authors correct on multiple comparisons?
  • Less patients with IPF received antibiotics. Please modify the text on page 7. Could this contribute to differences in mortality? I wonder if neutrophilia in the IPF group accompanied with the fact that they were less likely treated with antibiotics could have resulted in worse numbers.
  • The authors should mention when the lung function was performed before the AE.
  • There are numerous grammatical errors, and the authors should check the manuscript by a native speaker.
  • Please change “ILDs patients” to “patients with ILD” and “IPF patients” to “patients with IPF”.

Author Response

R1 General comments to Author

I have read the article by Faverio et al. with great interest. The authors compared IPF and non-IPF exacerbations of patients with ILD. This is a valuable work which should be endorsed for publication, should the authors address some comments.

R1.R

We thank the Reviewer for his/her consideration and appreciation of our manuscript.

R1.C1

I understand that this is a retrospective study, but post-hoc sensitivity analyses should be performed if the sample size was large enough to deliver the conclusions.

R1.R1

We thank the Reviewer for his/her comment. As stated in the paper “Aim of our study is to describe the patients’ characteristics at hospitalization for AE-ILD, comparing IPF with non-IPF patients, in two Italian ILDs referral Centers (University Hospitals of Monza and Modena), and to identify predictors of death at hospital discharge and of the overall survival.” Therefore, we are not focusing on a specific primary outcome and we cannot run a post-hoc analyses, because we do not have a unique hypothesis to test.

Furthermore, our study is exploratory in nature, and aims at detecting differences to give useful information for the planning of future studies.

R1.C2

Numerous comparisons were performed between the IPF and non-IPF groups. Did the authors correct on multiple comparisons?

R1.R2

We thank the Reviewer for his/her comment. We did not correct for multiple comparisons. Multiple comparison testing strategies are usually adopted for testing multiple primary outcomes, however the need for adjustment depends on the study nature. The paper “Li G, Taljaard M, et al. An introduction to multiplicity issues in clinical trials: the what, why, when and how. International journal of epidemiology, 2017; 46(2): 746-55” describes when such adjustments are needed. Since our study is exploratory, as stated in the previous comment, and since we are not comparing multiple outcomes, but rather the characteristics of two groups, we decided not to adjust.

R1.C3

Less patients with IPF received antibiotics. Please modify the text on page 7. Could this contribute to differences in mortality? I wonder if neutrophilia in the IPF group accompanied with the fact that they were less likely treated with antibiotics could have resulted in worse numbers.

 R1.R3

We thank the Reviewer for his/her comment. We changed the text as follows:

“During hospitalization for AE, patients were treated with different therapies according to clinical experience of our ILD team without any significant difference between the two groups, with the exception of antibiotic therapy, prescribed more frequently in patients with IPF”.

Active pulmonary infections on hospital admission were identified by the physician in charge based on patients’ signs and symptoms, laboratory and microbiological analysis results, and were excluded from the analysis (Figure 1). We used the 2007 definition of AE that excluded cases of AE triggered by pulmonary infection. In consideration of these inclusion/exclusion criteria, we do not think that the administration of antibiotic therapy itself may have had an impact on mortality. Patients with IPF were more likely to receive antibiotic therapy despite the absence of overt infection, and neutrophilia at exams performed on hospital admission may have contributed to this more frequent prescription.

R1.C4

The authors should mention when the lung function was performed before the AE.

R1.R4

We thank the Reviewer for this comment. We included pulmonary function tests performed within 12 months before the AE. We added this information in the Methods.

 R1.C5

There are numerous grammatical errors, and the authors should check the manuscript by a native speaker.

R1.R5

We thank the Reviewer for the suggestion. We reconsidered the paper and corrected the grammatical errors.

 R1.C6

Please change “ILDs patients” to “patients with ILD” and “IPF patients” to “patients with IPF”.

R1.R6

We thank the Reviewer for this suggestion. We changed the text as suggested.

Reviewer 2 Report

  1. Could you please add units wherever needed (example -- Neutrophilia value (Table 1), FVC - Median (Table 2)
  2. number of  digits should be checked for all parameters (example FEV1 % for non-IPF = 68, but for IPF = 78.775 )

Author Response

R2.C1

Could you please add units wherever needed (example -- Neutrophilia value (Table 1), FVC - Median (Table 2)

R2.R1

We thank the Reviewer for his/her suggestion. We added the units where needed.

R2.C2

number of digits should be checked for all parameters (example FEV1 % for non-IPF = 68, but for IPF = 78.775)

R2.R2

We thank the Reviewer for the comment. We corrected the Tables as suggested.